Diagnostic accuracy of fat pad sign, X-ray, and computed tomography in elbow trauma: implications for treatment choices—a retrospective study

Afacan Mustafa Ahmet drmustafaahmet@hotmail.com 1
Kilic Koray Kaya 2
Temiz Aytun 3
Tayfur İsmail 1
Doganay Fatih 1 4
1 Department of Emergency Medicine, University of Health Science , Istanbul , Turkey
2 Department of Radiology, Antalya Education and Research Hospital , Antalya , Turkey
3 Department of Ortopedic Surgery, Edremit State Hospital , Balıkesir , Turkey
4 Department of Research, Mayo Medical School , Jacksonville , FL , United States of America
Barak Meir
Electronic publication date: 2025 Feb 28
Publication date: 2025
Volume: 13
Electronic Location ID: e18922
Received 2024 Mar 27; Accepted 2025 Jan 10
Copyright: ©2025 Afacan et al.
Copyright year: 2025
Copyright holder: Afacan et al.
License: This is an open access article distributed under the terms of the Creative Commons Attribution License, which permits unrestricted use, distribution, reproduction and adaptation in any medium and for any purpose provided that it is properly attributed. For attribution, the original author(s), title, publication source (PeerJ) and either DOI or URL of the article must be cited.
License URL: https://creativecommons.org/licenses/by/4.0/

Keywords: Computed tomography, Elbow fracture, Elbow injury, Fracture, Surgery, Treatment, X-ray, Elbow trauma, Fat pad sign

Funding: The authors received no funding for this work.

==============================
Introduction

Identifying skeletal injuries significantly impacts patient outcomes in trauma cases. This study aims to compare the diagnostic accuracy of X-ray (XR) and computed tomography (CT) in detecting elbow fractures among patients presenting at the emergency department (ED). Additionally, the study assesses the potential contribution of the fat pad sign to enhancing the diagnostic accuracy of XR images in identifying elbow fractures. The secondary aim focused on evaluating the precision of XR imaging in determining the necessity for surgical intervention among patients presenting with elbow trauma.

Methods

Conducted retrospectively at an ED within a secondary hospital, this study included patients with elbow trauma between January 1, 2017, and January 1, 2020, who underwent both XR and CT imaging of the elbow joint. Following the application of exclusion criteria, the analysis comprised 183 patients based on remaining image data.

Results

When comparing XR to CT for fracture detection, XR exhibited a sensitivity of 46.9%, specificity of 85.9%, positive predictive value (PPV) of 79.3%, negative predictive value (NPV) of 58.4%, area under the curve (AUC) of 0.664. Considering the fat pad sign in XR as a fracture indicator, the sensitivity is 60.2%, specificity is 81.2%, PPV is 78.7%, NPV is 63.9% and AUC is 0.707. A significant difference was found when comparing the AUCs obtained with and without considering the fat pad sign (p = 0.039). Regarding surgical treatment decision-making, XR showed a sensitivity of 50%, specificity of 100%, PPV of 95%, NPV of 100%, and an AUC of 0.750 when compared to CT.

Conclusion

The findings indicate that XR alone is insufficient for detecting elbow fractures and determining the need for surgical treatment. Incorporating the fat pad sign improves the diagnostic accuracy of XR. In cases where suspicion of fracture is high, considering CT imaging is crucial to avoid missed diagnoses, prevent complications, and guide treatment decisions effectively.

Introduction

Elbow injuries are prevalent occurrences in Emergency Departments (ED), often resulting from sports-related traumas, accidents, or falls. Such injuries can significantly limit individuals’ daily activities and lead to both physical and psychological hardships (Yalçın et al., 2018; Chin, Chou & Peh, 2019).

Injuries to the elbow can encompass fractures, dislocations, and soft tissue disorders, typically assessed through X-ray (XR)—an initial imaging tool in ED settings (Griffith et al., 2001). The elbow is a complex joint formed by the interaction of the humerus, ulna, and radius. Owing to this intricate structure, diagnosing fractures solely through XR imaging in patients presenting with elbow trauma can sometimes prove challenging (Franklin et al., 1988; Savoie & O’Brien, 2017). While XR imaging is capable at detecting major joint fractures, its efficacy in identifying minor fractures may be limited, potentially resulting in lingering complications (Biz et al., 2019; Avci et al., 2019; Hussain, Siddique & Gillani, 2021).

In the assessment of fractures through XR, besides the direct appearance of fractures, indirect signs play a pivotal role. The distal humerus fossae contain adipose tissues, which can be displaced due to certain elbow injuries, leading to their displacement from the fossae (Bohrer, 1970). This displacement manifests as hypodensity visible on lateral radiographs in the anterior and posterior regions near the distal end of the humerus, recognized as “fat pad sign”. Typically, the fat pad sign is often associated with a radial head or supracondylar fracture (O’Dwyer et al., 2004).

Computed tomography (CT) imaging offers a more comprehensive evaluation of elbow injuries, providing detailed insights. However, it is a more expensive procedure compared to XR imaging and involves increased radiation exposure. As a result, CT imaging is not typically the initial diagnostic method for joint traumas. Instead, it is preferred after XR imaging when the clinician suspects a fracture, based on the incompatibility between XR findings and clinical signs or symptoms such as tenderness, swelling, or a limited range of motion (Walls, Hockberger & Gausche-Hill, 2017).

Prompt identification of skeletal injuries significantly enhances outcomes for patients presenting with trauma. Moreover, determining which patients should be referred to orthopedic surgeon stands as a crucial decision for ED physicians (Walls, Hockberger & Gausche-Hill, 2017).

The primary objective of this study was to compare the diagnostic efficacy of XR and CT in detecting bone fractures among patients presenting with elbow trauma. Additionally, the study aimed to assess the diagnostic precision of the fat pad sign. The secondary objective was to evaluate and compare the precision of XR and CT in guiding treatment decisions.

Materials & Methods

Study design

This retrospective study was conducted in the ED of a secondary hospital, following the principles of the Helsinki Declaration and after receiving approval from the local ethics committee (Balıkesir University Faculty of Medicine Ethics Committee, Ruling number: 2020/94).

A standardized data recording form was devised for the study, encompassing patient demographics, interpretations of both XR and CT images, as well as patient outcomes. Patient information was retrieved from the hospital database. Volunteerism or informed consent was not sought for this study, as it was conducted through a retrospective file review.

Patient selection

The study included individuals admitted to the ED due to elbow trauma between January 1, 2017, and January 1, 2020, whose elbow joint underwent imaging via both XR and CT. All age groups were considered eligible for inclusion. However, individuals meeting the following criteria were excluded from the study:

• Those who underwent imaging for non-traumatic reasons.

• Patients who had a plaster/splint applied before imaging.

• Cases where sagittal and coronal sections of CT images were unavailable.

• Patients whose elbow joint was incompletely scanned on either XR or CT.

• Patients who had undergone elbow arthroplasty.

• Cases involving internal or external fixators identified in any of the bones forming the elbow joint.

• Patients for whom XR or CT images were inaccessible for analysis.

XR imaging and CT scan

Patient XR and CT images were retrieved from the Hospital’s Picture Archiving and Communication System (PACS). Within the ED where this study took place, a Toshiba Alexion multi-slice CT scanner was utilized for CT imaging. Typically, extremity CT scans were reconstructed into three-dimensional reformatted images, including sagittal, coronal, and axial views. The XR imaging utilized the EMS-CMP 200 device with a U-arm HF radiographic system, presenting standard joint imaging in two planes: anterior-posterior (AP) and lateral.

Interpretation of images

The interpretation of both XR and CT images adhered to established methodologies from literature (Avci & Kozaci, 2019; Helms, 2013). To maintain objectivity, an experienced radiologist reinterpreted both XR and CT images randomly and in a blinded manner specifically for fracture analysis. Subsequently, an orthopedic surgeon conducted a blind review of all recorded XR images and their respective interpretations. This surgeon marked the digital dataset with the designation “surgery decision with XR”. Following this, the same orthopedic surgeon performed a blind evaluation of the CT images and their interpretations, marking the digital dataset with “surgery decision with CT”. Strict measures were implemented to ensure that the interpreting physicians did not have simultaneous access to both XR and CT images.

Statistical analysis

Statistical analysis utilized IBM SPSS Statistics 26.0 software (IBM Corp., Armonk, NY, USA). As the age data did not conform to a normal distribution, median values (interquartile ranges) were employed for its representation. Categorical data were compared using chi-square and Fisher’s Exact tests, expressed as frequency and percentage. In a chi-square analysis with an alpha level of 0.05, power of 0.80, sample size of 183, and degrees of freedom (DF) equal to five, the effect size reported is 0.26. The effect size of 0.26 reflects the strength of association between categorical variables in the analysis.

A receiver operating characteristic (ROC) analysis was performed employing the DeLong method to evaluate the diagnostic accuracy of the XR imaging method. This assessment was conducted both with and without considering the presence of the fat-pad sign in the analysis. Sensitivity, specificity, positive predictive value (PPV), negative predictive value (NPV), area under the curve (AUC), and Youden index J (YJI) values were calculated for the XR imaging method, assuming the CT imaging method as the gold standard (Zhou, 2011). YJI, ranging between zero and one, was utilized to evaluate diagnostic accuracy, where a value closer to one signifies increased test accuracy. Statistical significance was set at a p-value below 0.05. The null hypothesis posited no disparity between CT scan and XR imaging methods in diagnosing elbow joint injuries.

Results

Following the application of inclusion and exclusion criteria, the study encompassed 183 patients (Fig. 1). Of these, 101 patients were male, and 82 were female. The median age within the study population was 32 years (range: 12–52).

Figure 1 Study flow diagram.

Fractures were identified in 46 (25%) patients through XR imaging and in 98 (54%) patients through CT imaging. In comparison to CT, the sensitivity of XR for detecting fractures was 46.9%, while its specificity was 85.9%. PPV stood at 79.3%, and NPV at 58.4%. The AUC for XR imaging was calculated as 0.664, with a YJI of 0.33. When considering the fat pad sign in XR as indicative of a fracture, there were notable changes in accuracy indicators. Sensitivity increased from 46.9% to 60.2%, specificity decreased from 85.9% to 81.2%, PPV decreased from 79.3% to 78.7%, NPV increased from 58.4% to 63.9%, AUC improved from 0.664 to 0.707, and the YJI increased from 0.33 to 0.41. A statistically significant difference was observed when comparing the AUCs of XR with and without the fat pad sign (p = 0.039), as shown in Table 1. The fat pad sign was identified in XR images of 29 patients, among whom 25 were confirmed to have fractures on CT scans. However, of these 25 patients, XR successfully visualized fractures in only 12 cases.

When fractures were categorized based on their characteristics, XR imaging exhibited a specificity of 92% or higher for fissure, linear, spiral, fragmented, and avulsion-type fractures. Furthermore, specificities of 82% or higher were observed for angulation, stepping-off, extension of the fracture into the joint space, and growth plate fractures.

Sensitivity was observed to be 50% or lower across all mentioned fracture characteristics. Notably, 36% of the patients were under the age of 18, among whom two patients exhibited a growth plate fracture on XR imaging, while CT imaging detected growth plate fractures in 16 patients (Table 2). As shown in Table 2, when elbow trauma is assessed using only XR imaging, there is a statistically significant difference in identifying fractures with characteristics such as ‘Spiral (p = 0.046), Angulation (p = 0.002), and Extension to the joint space (p = 0.002),’ indicating that XR alone is inadequate for accurately detecting these fracture characteristics.

Table 1 The diagnostic accuracy of the XR with/without fat pad sign.

	CT Finding (+)	CT Finding (-)	Accuracy indicators	
	XR (+)	XR (-)	XR (+)	XR (-)	Sens.	Spes.	PPV	NPV	AUC*	p	YJI	
XR a	46 (46.9)	52(53.1)	12 (14.1)	73 (85.9)	46.9	85.9	79.3	58.4	0.664 (0.585–0.743)	<0.01	0.33	
XR and fat pad sign b	59 (61.5)	39(38.5)	16 (18.8)	69 (81.2)	60.2	81.2	78.7	63.9	0.707 (0.631–0.783)	<0.01	0.41	
Notes.

a The diagnostic accuracy of XR based on CT images with fracture detection.

b The diagnostic accuracy of XR when fat pad sign is considered as a fracture finding.

* When the AUC values of XR and XR+Sailing sign are compared, the p-value was calculated as 0.039.

XR X-ray

CT Computed tomography

Prev prevalence

Sens sensitivity

Spes specificity

PPV positive predictive value

NPV negative predictive value

AUC area under the curve

YJI Youden J index

Table 2 Comparison of XR and CT imaging findings based on the characteristics of the injury.

Characteristic	Prev.
%	Sens.
%	Spes.
%	PPV
%	NPV
%	AUC	p	
Fissure	13.1	8.33	92.45	14.3	87	0.504
(0.429–0.579)	0.898	
Linear	19.1	17.14	94.59	42.9	82.8	0.559
(0.484–0.632)	0.081	
Spiral	10.9	25	95.09	38.5	91.2	0.600
(0.526–0.672)	0.046	
Avulsion	0.55	0	98.4	0	99.4	0.492	0.977	
Fragmented	23	16.67	95.04	50.00	79.3	0.559
(0.483–0.632)	0.055	
Angulation	9.84	50	89.09	33.33	94.2	0.695
(0.623–0.761)	0.002	
Stepping-off	1.09	50	85.08	3.6	99.4	0.675
(0.602–0.743)	0.484	
Extension to the joint space	27.3	42	82.71	47.7	79.1	0.624
(0.549–0.694)	0.002	
Growth plate fracture	8.74	12.5	99.4	66.7	92.22	0.560
(0.484–0.633)	0.164	
Notes.

XR X-ray

CT Computed tomography

Prev prevalence

Sens sensitivity

Spes specificity

PPV positive predictive value

NPV negative predictive value

AUC area under the curve (confidence interval)

The total prevalence exceeds 100% as some cases involve multiple characteristics of injury.

Following the interpretation of CT images, surgical intervention was recommended for 18 (10%) patients. Among these cases, surgical treatment decisions consistent between XR and CT imaging in nine (5%) patients. However, for another group of nine (5%) patients, conservative treatment was suggested based on XR findings while surgical intervention was recommended upon CT assessment.

In determining the need for surgical intervention, the sensitivity of XR compared to CT imaging was calculated as 50%, with a specificity of 100%. Additionally, the PPV stood at 95%, and the NPV at 100%. The fracture characteristics of these 18 patients for whom surgical treatment was recommended are detailed in Table 3. As shown in Table 3, when surgical decisions for elbow trauma are based solely on XR imaging, XR is inadequate for accurately guiding decisions in fractures with characteristics such as linear (100%), spiral (42.9%), fragmented (53.9%), angulation (57.1%), stepping-off (50%), extension to the joint space (41.7%), and growth plate fractures (50%).

Table 3 Surgical decision prediction with XR in patients whose surgery was decided to be necessary with CT.

Characteristics of the injury on CT	XR (+)	XR(-)	Need for surgery total	
Fissure (n = 24)	1 (100%)	0 (0%)	1	
Linear (n = 35)	0 (0%)	3 (100%)	3	
Spiral (n = 20)	4 (57.1%)	3 (42.9%)	7	
Avulsion (n = 1)	0	0	0	
Fragmented (n = 42)	6 (46.1%)	7 (53.9%)	13	
Angulation (n = 18)	3 (42.9%)	4 (57.1%)	7	
Stepping-off (n = 2)	1 (50%)	1 (50%)	2	
Extension to the joint space (n = 50)	7 (58.3%)	5 (41.7%)	12	
Growth plate fracture (n = 16)	1 (50%)	1 (50%)	2	
All types	9 (50%)	9 (50%)	18	
Notes.

CT Computed Tomography

XR X-ray

Among the 183 participants, 25 had multiple characteristics of injury resulting in a total of 208 injuries.

The prevalence of all injuries exceeds 100% due to the presence of multiple characteristics of injury in some cases (Table 2). Among the 183 participants, 25 individuals had multiple injury characteristics, contributing to the total injury count of 208 (Table 3).

Discussion

This study showed how much CT imaging can impact surgical decisions for patients with elbow trauma who come to the emergency department. If only XRs are used and CT scans are not done, half of the patients with elbow injuries needing surgery may not be properly diagnosed or treated. Accurately diagnosing fractures in the elbow poses a significant challenge when relying on XR imaging only, owing to the intricate nature of the elbow’s anatomy. Diagnostic evaluations for such injuries typically encompass various imaging techniques, including standard radiographs, cross-sectional imaging, and dynamic ultrasonography (Chin, Chou & Peh, 2019; Avcıet al., 2016).

Our study primarily aimed to compare the diagnostic accuracy of XR with CT in detecting elbow bone fractures. The sensitivity of XR in diagnosing these fractures within our patient cohort was observed to be less than 50%. This finding signifies a substantial limitation as more than half of the fractures went undetected by XR, highlighting its restricted ability to identify such injuries. These limitations emphasize the challenge faced by XR in capturing bone fractures associated with elbow trauma.

An important result in our study was the identification of the fat pad sign’s efficacy during lateral XR imaging of the elbow. This sign indicates the presence of intra-articular elbow effusion or hemarthrosis and has consistently been linked to intra-articular elbow injuries or fractures. Studies have reported that fat pad sign significantly enhances XR’s diagnostic accuracy in detecting fractures, even when standard elbow radiographs fail to visibly display a fracture (Bohrer, 1970; O’Dwyer et al., 2004; Al-Aubaidi & Torfing, 2012; De Froda et al., 2017; Poppelaars et al., 2022; Hussein et al., 2023). Our investigation found fat-pad sign in XR images of 29 patients, among whom 25 exhibited fractures detected by CT scans. Remarkably, XR only identified fractures in 12 of these cases. In our study, considering fat pad sign improved XR’s ability to detect fractures, indicating its potential as an indicator for more sensitive assessment.

Moving beyond the diagnosing of fracture, specific fracture characteristics such as location, type, and associated features crucially influence treatment decisions for elbow bone fractures (Sheehan et al., 2013; Nocerino et al., 2018). CT imaging offers advantages in detecting hidden fractures, elucidating displaced fracture origins within the joint, and offering a clearer depiction of joint anatomy and fracture morphology compared to XR (Griffith et al., 2001; Avcıet al., 2016; Haapamaki, Kiuru & Koskinen, 2004; Lubberts et al., 2016; Mellema et al., 2015; Ricci et al., 2019). In our investigation, XR’s sensitivity consistently remained at 50% or lower across various fracture characteristics, including stepping-off, angulation, extension into the joint space, and growth plate fractures. Our study illustrates lower sensitivity with XR compared to CT, highlighting CT’s ability to provide additional crucial information on elbow fractures.

These findings show the importance of preferring CT imaging in suspected elbow fractures, especially when precise evaluation of specific fracture characteristics is necessary for optimal treatment decisions. Despite its disadvantages in terms of increased cost and radiation exposure, CT imaging stands out for its ability to provide a more detailed and accurate assessment, critical for deciding the appropriate treatment methods in cases of elbow trauma.

Studies comparing the accuracy of XR and CT imaging in joint injuries consistently demonstrate the lower efficacy of XR. They reveal that XR often fails to provide sufficient information compared to CT scans in various joint traumas, leading to significant alterations in diagnosis and treatment approaches (Franklin et al., 1988; Avci et al., 2019; Ricci et al., 2019; Etli et al., 2020; Al-Ani et al., 2022).

Preoperative CT scans have shown substantial influence in managing diverse joint traumas, often resulting in significant alterations in surgical plans, techniques, and approaches. Exclusively relying on XR imaging has been insufficient for adequately defining fracture characteristics (Chen et al., 2015; Gibson et al., 2017; Tornetta & Gorup, 1996; Martinelli et al., 2024; Nguyen et al., 2023).

In our study, the decision for surgical intervention increased to 10% following CT imaging, compared to a 5% decision rate when only XR images were available. When evaluating the decision-making process for surgical treatment, XR demonstrated a sensitivity of 50%. This findings indicate the inadequacy of XR imaging in guiding surgical treatment decisions. Our study stands as pioneering research, highlighting the effectiveness of CT imaging in guiding surgical decisions concerning elbow fractures. Our results emphasize the significance of CT imaging, aligning with prior studies conducted on fractures in other joints, and demonstrating its crucial role in treatment decisions.

One of the key limitations of our study is its retrospective nature, which inherently limits our ability to control for all potential confounders and biases. Our study’s scope is also limited due to its single-center design and the relatively small sample size. To obtain more comprehensive and generalizable results, further prospective studies involving larger, multi-center populations are necessary. Conducting prospective, larger-scale, multi-center studies will significantly contribute to enhancing our understanding of the diagnostic accuracy of imaging modalities in elbow traumas and their impact on treatment decisions.

Conclusions

XR remains the primary imaging modality in cases of extremity traumas. However, its limitations in accurately detecting the presence of fractures, determining fracture types and characteristics, as well as guiding surgical treatment decisions in elbow traumas have been underscored. Observing the presence of the fat pad sign could potentially improve the diagnostic accuracy of XR when fractures are not visibly detected in elbow XR. Especially in cases with a high suspicion of a fracture, CT imaging becomes necessary to avoid missing fractures, prevent complications, and enable accurate treatment decisions.

Supplemental Information

Supplemental Information 1 Raw data

Supplemental Information 2 Codebook

Additional Information and Declarations

Competing Interests

Author Contributions

Human Ethics

Data Availability

The authors declare there are no competing interests.

Mustafa Ahmet Afacan conceived and designed the experiments, performed the experiments, analyzed the data, prepared figures and/or tables, authored or reviewed drafts of the article, and approved the final draft.

Koray Kaya Kilic performed the experiments, analyzed the data, authored or reviewed drafts of the article, and approved the final draft.

Aytun Temiz performed the experiments, analyzed the data, authored or reviewed drafts of the article, and approved the final draft.

İsmail Tayfur analyzed the data, authored or reviewed drafts of the article, and approved the final draft.

Fatih Doganay conceived and designed the experiments, performed the experiments, analyzed the data, prepared figures and/or tables, authored or reviewed drafts of the article, and approved the final draft.

The following information was supplied relating to ethical approvals (i.e., approving body and any reference numbers):

Balıkesir University Faculty of Medicine Ethics Committee (number: 2020/94).

The following information was supplied regarding data availability:

The raw data are available in the Supplementary File.

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
