# Peer review of "Diagnostic accuracy of fat pad sign, X-ray, and computed tomography in elbow trauma: implications for treatment choices—a retrospective study"

_PeerJ, doi:10.7717/peerj.18922_

## Round 0.1 · original submission · Major Revisions

Dear Dr. Afacan,

Your manuscript titled "Diagnostic accuracy of X-Ray imaging in elbow trauma: Implications for treatment choice" was considered by two expert reviewers and based on their opinions and my review, the decision is “Major revision”.

Please carefully read reviewer 2 comments. The reviewer correctly noted that the manuscript presents information and conclusions that are already well known – mainly, that incorporating the fat pad sign to x-rays data will improve the accuracy of elbow fractures diagnostic). Thus, the originality of this study, as it is currently presented, is limited at best.

If the authors believe they can revise the manuscript in a way that will bring some novelty and new insights to the topic, they are encouraged to resubmit.

When you resubmit, please ensure that all review, editorial, and staff comments are addressed in a response letter and any edits or clarifications mentioned in the letter are also inserted into the revised manuscript where appropriate.

Please note that submitting a revision of your manuscript does not guarantee eventual acceptance, and that your revision may be subject to re-review by the reviewer(s) before a decision is rendered.

Sincerely,
Meir Barak PhD, DVM

Reviewer 1 ·

Basic reporting

Clear and unambiguous, professional English used throughout.


Literature references, sufficient field background/context provided.


Professional article structure, figures, tables. Raw data shared.


Self-contained with relevant results to hypotheses.

Experimental design

Research question well defined, relevant & meaningful.


It is stated how research fills an identified knowledge gap.


Rigorous investigation performed to a high technical & ethical standard.


Methods described with sufficient detail & information to replicate.

Validity of the findings

All underlying data have been provided; they are robust, statistically sound, & controlled.

Conclusions are well stated, linked to original research question & limited to supporting results.

Additional comments

The authors investigated performance of prediction of X-Ray with and without fat pat sign in elbow fractures. Also they showed that need of surgery .
Reliability of materials, methods, and results, With the literature cited, the discussion is extensive.
X-Ray is the basic imaging method which frequently used in the emergency department for trauma. The results of the study revealed that observing the presence of the fat pad sign could potentially improve the diagnostic accuracy of XR when fractures are not visibly detected in elbow XR.
In contrast to a high suspicion of a fracture, CT imaging was showed that necessary to avoid missing fractures and enable accurate treatment decisions.
The authors confirmed the hypothesis according to this results. Accept without corrections.

·

Basic reporting

The manuscript titled "Diagnostic Accuracy of X-Ray Imaging in Elbow Trauma: Implications for Treatment Choice" is well-written and uses clear, professional English throughout. The introduction provides sufficient context, and the background is well-referenced and relevant. The manuscript structure conforms to PeerJ standards, and the figures are of high quality, well-labeled, and described. The raw data is supplied as per PeerJ policy. However, the article lacks novelty, as it mainly reiterates findings that are already well-documented in the existing literature.

Experimental design

The study is a retrospective analysis comparing the diagnostic accuracy of X-ray and CT in detecting elbow fractures. The research question is relevant and well-defined, but it does not fill a significant knowledge gap. The investigation is performed to a high technical and ethical standard, and the methods are described with sufficient detail to replicate. However, the study does not introduce any new methods or approaches, which limits its originality and impact.

Validity of the findings

The findings indicate that X-ray has low sensitivity and specificity in diagnosing elbow fractures and that incorporating the fat pad sign improves diagnostic accuracy. While the data is robust and statistically sound, the conclusions drawn are not novel. Similar results have been extensively reported in the literature, and the study does not offer new insights or breakthroughs. The manuscript lacks a discussion on how these findings advance the field beyond what is already known.

Additional comments

The manuscript is well-executed in terms of study design and data analysis. However, it falls short in terms of innovation. The conclusions drawn are consistent with existing knowledge and do not significantly contribute to the advancement of the field. For these reasons, I recommend rejecting the manuscript. The authors are encouraged to focus on more innovative and forward-looking topics in their future research.

---

## Round 0.2 · Minor Revisions

Dear Dr. Afacan,

Your revised manuscript titled "Diagnostic accuracy of X-Ray imaging in elbow trauma: Implications for treatment choice" was further considered by two expert reviewers and based on their opinions and my review, the decision is “Minor revision”.

Please carefully read reviewer 4 comments. Among other points, they recommend adjusting the title, adding some references and clarifying the limiting factors of your study. Please note that you *don’t* need to delete all references older than the year 2000, but it is recommended that you should add some additional more recent references.

In addition, please address the following points:
(1) L121-6 (Results): This section should be moved to the materials and methods section.
(2) Table 2 – Prevalence. Shouldn’t the prevalence of all injuries sum up to 100%? Currently I get a value of 113.62%. Where there some cases involving multiple fractures? From table 3 it looks like total injuries = 208, which is more than the number of participants (183), so it seems to be the case. This should be clearly indicated in the table and text.
(3) In your response to reviewer 2, regarding the novelty of your study, you have stated that this study does contribute a significant addition to the existing literature as it specific “within our defined patient population”. Can you please explain what you mean.
(4) Please convert the supplementary file “peerj-96382-Elbow_dataset_Peerj_v3” from an SPSS file (.sav) to a file type that can be open with a such specific software (e.g., PDF, RTF, Word, Excel etc,).

When you resubmit, please ensure that all review, editorial, and staff comments are addressed in a response letter and any edits or clarifications mentioned in the letter are also inserted into the revised manuscript where appropriate.

Please note that submitting a revision of your manuscript does not guarantee eventual acceptance, and that your revision may be subject to re-review by the reviewer(s) before a decision is rendered.

Sincerely,
Meir Barak PhD, DVM

Reviewer 3 ·

Basic reporting

no comment

Experimental design

no comment

Validity of the findings

no comment

Additional comments

Dear authors,

Thank you for the opportunity to review your manuscript. I appreciate the considerable effort you have put into addressing the diagnostic challenges in elbow trauma. Your study provides valuable insights into the comparative effectiveness of X-ray and CT imaging, particularly emphasizing the utility of the fat pad sign in enhancing X-ray diagnostic accuracy.

Your findings contribute significantly to our understanding of imaging modalities in emergency settings and offer practical guidance that could influence clinical decision-making processes. However, to further strengthen your manuscript, I would recommend elaborating on the specific clinical scenarios where CT should be prioritized over X-ray, based on the severity or type of fractures observed. Additionally, providing more detailed case studies or examples where your findings have directly influenced patient outcomes could enrich the context and practical relevance of your research.

Overall, your work is commendable, and with some additional details, it could serve as a robust resource for both clinical practice and academic research. I look forward to seeing these enhancements in your revised manuscript.

Best regards,

Reviewer 4 ·

Basic reporting

Review

Many thanks to the authors for having presented a so interesting retrospective study about “Diagnostic accuracy of X-Ray imaging in elbow trauma: implications for treatment choice - A retrospective study”.

Before resubmitting the revision version of the article, please read the editorial rules carefully, and check other editorial aspects, such as: text alignment (lacking), text justification at the head (lacking), etc. The language is no good, hence the manuscript needs to be corrected by a person of
English mother tongue.

Title and Abstract
The title is too poor and does not capture the true meaning of the article, please insert in the title text references to computed tomography.
The abstract is well structured, and it contains the main information of the study.

Background
The introduction is quite well structured, containing the main aims of the study.
Line 41: elbow injuries often resulting from sport-related traumas. What kind of sport? Is there a type of sport that increases the incidence of elbow trauma?
Line 45: ‘The elbow is a complex joint formed by the interaction of three bones’: Which bones? It might be interesting to add information about the injury mechanism. After elbow injuries a dislocation can be possible with following simplex or complex instability.

Line 54: ‘Typically, the anterior fat pad sign is often associated with radial head or supracondylar fracture’: and the posterior? Extend the argument on the fat pad sign.
Line 59: ‘it is preferred following XR imaging, based on the clinician’s suspicion’: what is the sign that can suggest an unknown fracture?


Results
The result presented are quite complete, reflecting the MM section.
Here too, possible confounding factors are not taken into account.


References
The references are up to date, but they should be integrated as suggested.
Please delete references before the year 2000.

Competing interest
The authors declare that they have no financial or personal interest.

Tables and Figures
The number and quality of tables are appropriate to transmit the main information of the paper.
There are no figures. Please add it.

Experimental design

Methods
This section contains enough information to understand and possibly repeat the study.
However, no confounding factors or bias are indicated.

Statistical analysis
The statistical analysis is appropriate to the research.
Howerever, the statistical methods used to control confounding factor are not taken in to account.
It is not indicated who performed the analysis; please, be specific.

Validity of the findings

Discussion
The length and content of the discussion communicates the main information of the paper.
Any other recent work on the subject is not considered.
The limitations of the manuscript are only referred to the numerical sample when other factors could be taken into account.

Conclusions
The conclusions only reflect and refer to the results of the study.

Additional comments

None

---

## Round 0.3 · Minor Revisions

Dear Dr. Afacan,

Your revised manuscript titled "Diagnostic accuracy of X-Ray imaging in elbow trauma: Implications for treatment choice" was further considered by an expert reviewer and based on their opinion and my review, the decision is “Minor revision”.

Please read reviewer 4 comments. Their comment for line 45 requests that you add another reference. Please read the relevant reference and decide if it is suitable or not. I will accept your decision on the matter.

When you resubmit, please ensure that all review, editorial, and staff comments are addressed in a response letter and any edits or clarifications mentioned in the letter are also inserted into the revised manuscript where appropriate.

Please note that submitting a revision of your manuscript does not guarantee eventual acceptance, and that your revision may be subject to re-review by the reviewer(s) before a decision is rendered.

Sincerely,
Meir Barak PhD, DVM

Reviewer 4 ·

Basic reporting

The following point was not addressed as the reference was removed from my original revision.
Please, provide it.
Thank you!

Background
The introduction is quite well structured, containing the main aims of the study.
Line 41: elbow injuries often resulting from sport-related traumas. What kind of sport? Is there a type of sport that increases the incidence of elbow trauma?
Line 45: ‘The elbow is a complex joint formed by the interaction of three bones’: Which bones? It might be interesting to add information about the injury mechanism. After elbow injuries a dislocation can be possible with following simplex or complex instability. Please, add these aspects quoting:
https://pubmed.ncbi.nlm.nih.gov/31773896/

Experimental design

OK

Validity of the findings

OK

Additional comments

None

---

## Round 0.4 · accepted · Accept

Dear Dr. Afacan,

After careful consideration of your revisions, we have determined that your work meets our publication standards. Thus, I am pleased to inform you that your revised manuscript, "Diagnostic accuracy of X-Ray imaging in elbow trauma: Implications for treatment choice", has been accepted for publication.